# Hyperpolarized Dihydroxyacetone Is a Sensitive Probe of Hepatic Gluconeogenic State

**DOI:** 10.3390/metabo11070441

**Published:** 2021-07-05

**Authors:** Mukundan Ragavan, Marc A. McLeod, Anthony G. Giacalone, Matthew E. Merritt

**Affiliations:** Department of Biochemistry and Molecular Biology, University of Florida, Gainesville, FL 32610, USA; mukundan@ufl.edu (M.R.); marc.mcleod@ufl.edu (M.A.M.); anthonygiacalone@ufl.edu (A.G.G.)

**Keywords:** hyperpolarization, hepatic gluconeogenesis, dynamic nuclear polarization, diabetic liver

## Abstract

Type II diabetes and pre-diabetes are widely prevalent among adults. Elevated serum glucose levels are commonly treated by targeting hepatic gluconeogenesis for downregulation. However, direct measurement of hepatic gluconeogenic capacity is accomplished only via tracer metabolism approaches that rely on multiple assumptions, and are clinically intractable due to expense and time needed for the studies. We previously introduced hyperpolarized (HP) [2-^13^C]dihydroxyacetone (DHA) as a sensitive detector of gluconeogenic potential, and showed that feeding and fasting produced robust changes in the ratio of detected hexoses (6C) to trioses (3C) in the perfused liver. To confirm that this ratio is robust in the setting of treatment and hormonal control, we used ex vivo perfused mouse livers from BLKS mice (glucagon treated and metformin treated), and *db/db* mice. We confirm that the ratio of signal intensities of 6C to 3C in ^13^C nuclear magnetic resonance spectra post HP DHA administration is sensitive to hepatic gluconeogenic state. This method is directly applicable in vivo and can be implemented with existing technologies without the need for substantial modifications.

## 1. Introduction

Diabetes is a widely prevalent ailment affecting more than 10% of adults in the United States, with about 20% of those undiagnosed. A further 34% of the adult population are pre-diabetic based on fasting glucose or A1C levels [1]. Among the existing treatment paradigms for diabetes, targeting hepatic gluconeogenesis using the widely prescribed pharmaceutical metformin is the most common strategy. However, the inability to directly measure hepatic gluconeogenesis has hampered the development of alternative treatments and leaves the clinician uncertain of treatment efficacy at a mechanistic level.

Hyperpolarization via Dynamic Nuclear Polarization makes ^13^C magnetic resonance (MR) imaging tractable, without the need for signal averaging [2]. Hyperpolarization enables real-time studies of in vitro chemical reactions [3,4], protein folding [5,6] and interactions [7], and metabolism [8]. Over the last decade, hyperpolarized substrates have been utilized to study metabolism using cell culture [9,10], ex vivo perfusions (for example, heart and liver) [11,12,13,14] and in whole animals [15,16,17,18]. Hyperpolarized (HP) pyruvate has also been used to monitor cardiac metabolism [19,20], brain metabolism [21,22], and prostate cancer [23,24] in human patients. Several different substrates [8] have been used in the context of HP studies of metabolism. While pyruvic acid is the most common substrate, we have previously shown that imaging with [2-^13^C]dihydroxyacetone provides unique insights into hepatic physiology [25].

Dihydroxyacetone (DHA) is a small molecule that is a precursor of dihydroxyacetone phosphate (DHAP), which is ideally located at the mid-point of the Embden-Meyerhoff pathway. DHA is rapidly absorbed by tissues upon administration and is immediately phosphorylated into DHAP and further metabolized to three possible fates: towards glucose, to glycerol via glycerol-3-phosphate, or towards pyruvate, and the TCA cycle. DHA is a potent gluconeogenic substrate, exceeding the capacity of the majority of 3-carbon intermediates [26]. HP DHA has already been used to study basal hepatic metabolism [25] and acute hepatic and renal metabolic changes induced by glucose/fructose challenge [27,28]. Based on previous studies where we showed that HP DHA is sensitive to changes in hepatic gluconeogenesis between fed and fasted states, we hypothesized that DHA will be a uniquely sensitive probe to monitor perturbations in hepatic gluconeogenesis. To test this hypothesis, we use three study groups-pharmaceutical manipulation using metformin, hormonal manipulation using glucagon, and a disease state of diabetes, which is approximated here using the leptin receptor deficient *db/db* mouse model. It has been shown that *db/db* mice have large livers, develop loss of insulin sensitivity, and exhibit elevated hepatic gluconeogenesis, making these mice a good model of diabetes [29].

## 2. Results

Livers from fasted C57BLKS/J (henceforth referred to as “BLKS”) and diabetic (*db/db*) mice were perfused via the portal vein prior to hyperpolarized dihydroxyacetone exposure. Livers from BLKS mice were perfused with either metformin or glucagon as detailed in Methods (Section 4.1). Figure 1a,b compares liver mass and mouse total body mass between the three groups (*db/db,* glucagon, and metformin). As expected, *db/db* mice show increased body and liver mass as compared to BLKS mice, while there was no difference in mice and liver mass used in glucagon and metformin groups. Perfused liver function was determined by oxygen consumption. Hepatic oxygen consumption during the perfusions, normalized for liver mass, was not different between the BLKS livers (either group) and *db/db* livers, as shown in Figure 1c.

Hyperpolarized [2-^13^C]DHA (solid-state polarization achieved is 30 ± 9%; solution state enhancement is ~15,000) is rapidly converted to several metabolites post administration, as shown in Figure 2. In the metformin perfused livers, ^13^C signal intensities corresponding to three-carbon intermediates such as glycerol, glycerol-3-phosphate, and glyceraldehye-3-phoshate were predominant (Figure 2 bottom trace). In the glucagon perfused livers, however, ^13^C signals corresponding to several six-carbon and three-carbon metabolites were present (Figure 2 middle trace). It would be expected that the *db/db* livers show increased glucose production (and consequently, higher intensities of six-carbon signals). Indeed, HP DHA administration to the *db/db* livers resulted in ^13^C spectra that had resonances belonging to several hexoses and hexose phosphates in addition to three-carbon intermediates.

In order to ascertain the sources of glucose production in the liver post HP DHA administration, Gas Chromatography—Mass Spectrometry (GC-MS) analysis of glucose extracted from the liver was carried out. Using aldonitrile pentapropionate derivatization [30,31], glucose fragments containing carbons 1–3 (C1–3 215–220 *m*/*z*) and 4–6 (C4–6 259–264 *m*/*z*) were obtained (Figure 3). Since mice were fasted overnight, de novo gluconeogenesis should be primarily responsible for hepatic glucose production in these perfused livers. On average, livers from *db/db* mice produced ~158 micromoles of glucose per gram liver, while BLKS mice in glucagon and metformin groups produced ~23 and ~8 micromoles (per gram liver), respectively (Appendix A).

The total enrichment of glucose produced by *db/db* livers is lower than in other manipulations, as shown in Figure 3, likely due to much higher amounts of glycogen (lower depletion) in the diabetic liver. There were no differences in the total enrichment between the two fragments (C1–3 vs. C4–6) between glucagon and metformin groups. In the *db/db* livers, M+1 enrichment in the C1-3 fragment was more than 3× higher than that of the C4–6 fragment (Figure 3c; *p* = 0.01), suggesting incomplete triose phosphate isomerase equilibration or differential ^2^H incorporation. Glucose produced via [U-^13^C]propionate anaplerosis was more limited, as evidenced by low amounts of larger mass (M+2, M+3, M+4) isotopomers (Figure 3).

Based on the ^13^C HP spectra, it is possible to arrive at a simple metric that distinguishes the high gluconeogenic states (such as in diabetes or glucagon administration) from non-gluconeogenic livers using HP DHA as a substrate. The ratio of integrals of hexose resonances to three- and four-carbon metabolites derived from HP DHA (identified in Figure 2) is sensitive to de novo hepatic glucose production. This “6C/3C ratio” is more than three times larger in *db/db* (0.56 ± 0.24) and glucagon treated livers (0.53 ± 0.22) compared to metformin-treated livers (0.15 ± 0.14), as shown in Figure 4. A comparison of 6C/3C ratios from these three groups to the 6C/3C ratio from untreated BLKS livers is shown in Appendix A.

## 3. Discussion

An established system for studying type II diabetes is the leptin receptor deficient *db/db* mouse model. Livers from *db/db* are almost twice as large as livers from the control BLKS mice. Since *db/db* mice livers have been shown to contain higher amounts of lipid and glycogen but similar amounts of total protein [29], it was expected that the hepatic oxygen consumption, normalized to liver mass, would be similar (Figure 1). Similarly, the elevated total glucose output from *db/db* livers is also consistent with available literature [29].

Endogenous glucose production (EGP) was substantially different between diabetic and non-diabetic mice as well (Appendix A). Livers from *db/db* mice produced, on average, greater than five times glucose compared to livers from BLKS mice (both groups). Although glucagon-treated mice produced more glucose, it was not statistically different from metformin group due to greater variability of EGP in the glucagon group. Nevertheless, metabolism of hyperpolarized DHA reflects the profound differences in hepatic gluconeogenesis in diabetic and non-diabetic mice (Figure 2), even when the differences in EGP are not readily apparent.

For complete analysis of the data, the two distinct segments of the study should be considered—(i) liver perfusion prior to HP DHA administration and (ii) post HP DHA administration. Analysis of freeze clamped liver tissue provides the most accurate picture of DHA metabolism while analysis of the perfusate primarily yields information about liver metabolism prior to DHA administration, as it essentially “integrates” for metabolism over the full perfusion length. The observations from the HP experiments are validated by GC-MS measurements of glucose enrichment in the liver. GC-MS measured glucose enrichment in the liver yields insight on glucose production during the period of DHA administration and for the remainder of the perfusion. All three groups show higher enrichment of M+1 species, indicating glucose production primarily from hyperpolarized DHA (Figure 3). Although D_2_O is present in the perfusate (see Materials and Methods), the amount of deuterium label available is reduced due to injection of hyperpolarized substrate in a non-deuterated solvent. The volume of HP DHA injection (~23 mL; see Materials and Methods) dilutes the perfusate in the perfusion column by a factor of two. Therefore, the maximal enrichment possible from D_2_O during and after the HP DHA injection is ~5%. Although *db/db* M+1 enrichment is around that value, glucose produced from a single molecule of [2-^13^C]DHA (and no deuterium incorporation) will also produce M+1 mass isotopomer. Considering that the HP spectra from *db/db* livers show, in real time, glucose production, the observed M+1 enrichment points to DHA being a major contributor to glucose enrichment following the injection.

Similar analysis of perfusate derived glucose characterizes glucose production during the perfusion prior to HP DHA administration (Appendix A). As expected, total enrichment of glucose from *db/db* livers was significantly higher in the perfusate, as measured from C1–3 and C4–6 fragments (Appendix A). Glucose produced by livers in the glucagon group was not significantly different between liver and perfusate possibly due to different levels of glycogen still present in the livers even after fasting. Furthermore, since glucagon differentially promotes different substrates for gluconeogenesis [26], the presence of unlabeled lactate and pyruvate in the perfusate may have contributed to lower glucose enrichment. However, M+2 and M+3 glucose enrichment was higher in C4–6 fragments for metformin-treated livers, suggesting increased usage of propionate (plus deuterium incorporation) as a substrate for gluconeogenesis. This is likely further facilitated by the putative inhibition of glycogenolysis by metformin [32].

In order to assess hepatic energetics and establish ^13^C label propagation from propionate, we analyzed pool sizes and ^13^C enrichment of several citric acid cycle intermediates, as shown in Appendix A. No pool size differences were obtained between the three groups among either the citric acid cycle intermediates or lactate and alanine (Appendix A). However, there were differences in total enrichment (^13^C and ^2^H) in G3P and succinate. In *db/db* livers, the higher M+1 mass isotopomer of G3P (Appendix A) suggests higher DHA utilization. Similarly, the almost 10% enrichment of M+3 mass isotopomer in succinate shows label incorporation from propionate in the citric acid cycle. Overall, substrate utilization and hepatic energetics were similar between the livers in the three groups, as evidenced by these results and the O_2_ consumption.

The minimal differences in pool sizes of citric acid cycle intermediates and closely associated metabolites (such as aspartate and glutamate) between diabetic livers and non-diabetic livers is unsurprising, as this matches previous observations [33,34]. The C6/C3 ratio proposed here as a measure of gluconeogenesis is a simple measure of hepatic gluconeogenesis and requires few additional parameters for normalization since each measurement is normalized internally. This recapitulates data previously observed in the fed versus fasted liver, and shows that HP [2-^13^C]dihydroxyacetone is acutely sensitive to any source modulating gluconeogenic output. If necessary, kinetic models can be utilized to describe the evolution of HP signals arising from DHA metabolism to extract kinetic information [35]. Multiple studies in vivo in the rat suggest that DHA can serve as an agent for human imaging [27,28,36].

Regulating hepatic gluconeogenesis is an important target for treating Type II diabetes and is an important co-variate in the emerging pathology of nonalcoholic steatohepatitis. Numerous methods are available to measure hepatic glucose output and have been evaluated in detail elsewhere [37]. The most widely used technique to measure gluconeogenesis is by measuring deuterium incorporation into glucose from body water enrichment (oral D_2_O administration to reach ~0.5% body water enrichment). Blood glucose is then analyzed using NMR spectroscopy or mass spectrometry. In contrast, the method described in this work is straightforward since very little sample processing or extensive data analysis is required. Hyperpolarized techniques have been shown to accurately reflect the metabolic state of various tissues under in vivo and ex vivo modalities. In this study, we have demonstrated that hyperpolarized DHA is sensitive to modulations of hepatic glucose output and propose a simple metric to distinguish gluconeogenic and non-gluconeogenic livers.

## 4. Materials and Methods

### 4.1. Liver Perfusions

Experiments involving mice were handled in compliance with the University of Florida Institutional Animal Care and Use Committee (protocol number #201909320). Strains used were C57BLKS/J (Stock No. 000662) and BKS.Cg-Dock7^m^+/+Lepr^db^J (*db/db*). C57BLKS/J mice are the appropriate background control for the *db/db* strain. All male mice were 10–13 weeks old. Mice were anesthetized using isoflurane followed by an intraperitoneal injection of heparin. About 10 min after heparin injection, a celiotomy was performed under anesthesia (lidocaine was administered subcutaneously prior to making the first incision) exposing the liver and the portal vein. The portal vein was then cannulated and perfusion was started. The liver is then excised from the body and connected to a glass perfusion column. This glass perfusion apparatus is then moved into the bore of an NMR magnet.

Livers were perfused with perfusate containing Krebs-Henseleit electrolytes (25 mM NaHCO_3_, 112 mM NaCl, 4.7 mM KCl, 1.2 mM each of MgSO_4_ and KH_2_PO_4_ and 0.5 mM sodium-EDTA, 1.25 mM CaCl_2_), 6 mM sodium lactate, 0.6 mM sodium pyruvate, 0.2 mM [U–^13^C] sodium propionate, 10% (*v*/*v*) D_2_O, and 0.63 mM mixed fatty acids (containing palmitic acid (22.1% of total), palmitoleic acid (5.2%), stearic acid (2.7%), oleic acid (27%), linoleic acid (37.7%), γ-linolenic acid (2.4%), and decosahexanoic acid (2.8%)) along with 2% (*w*/*v*) bovine serum albumin. For perfusions with livers from BLKS mice, either 10 mM metformin hydrochloride or 50 nM glucagon was also added. Perfusate was oxygenated using 95% O2/5% CO_2_ mixed gas for the duration of perfusion. Oxygen consumed by the liver during perfusion was measured every 15 min using an Oxygraph+ setup (Hansatech Instruments, Norfolk, UK). At the end of the perfusion, the livers were freeze-clamped using liquid nitrogen and stored at −80 °C until needed. The number of mice used in each group was *db/db* (n = 6), glucagon (n = 5), and metformin (n = 6).

### 4.2. Dynamic Nuclear Polarization

[2–^13^C] dihydroxyacetone dimer (Sigma Isotec, St. Louis, MO, USA) was dissolved to a final concentration of 4 M in a 2:1 mixture of dimethyl sulphoxide and water doped with 15 mM trtiyl radical (tris[8-carboxyl-2,2,6,6-tetra-[2-(1-hydroxyethyl)]-benzo-(1,2-d:4,5-d)-bis-(1,3)-dithiole-4-yl]-methyl sodium salt; Oxford Instruments, Abingdon, UK) and 1 mM ProHance. This sample was inserted into HyperSense polarizer (Oxford Instruments, Abingdon, UK). The frozen sample was hyperpolarized at 1.4 K by applying microwave irradiation (94.116 GHz at 100 mW) until steady state was achieved (~1.5–2 h). The hyperpolarized sample was rapidly dissolved using 4 mL of hot 20 mM PBS (pH 7.4) and transferred to a 14 T NMR magnet. 3 mL of the dissolved sample was mixed with 20 mL of oxygenated Krebs-Henseleit electrolytes and injected directly into the perfused liver (nominal final concentration of HP DHA dimer is 4 mM). NMR signal acquisition was started prior to injection.

### 4.3. NMR Spectroscopy

Spectra in hyperpolarized DHA experiments were collected in a custom-built 20 mm broadband probe (Qonetec, Dietlikon, Switzerland) installed in a 14 T magnet equipped with an Avance III NMR console (Bruker Biospin, Billerica, MA, USA). Prior to injection of HP DHA, shimming was carried out using the ^23^Na signal. Typically, linewidths of around 18 Hz in ^23^Na spectrum were achieved (translating to similar linewidths in ^13^C spectra). ^13^C spectra (spectral width of 424 ppm; 12,280 data points) were recorded using a 45° {x, −x} binomial excitation scheme centered on DHA resonance (nominally, 212 ppm) with ^1^H decoupling (WALTZ65; B1 = 4.5 KHz) during acquisition and a repetition time of 3 s. Spectra were processed with 15–20 Hz exponential line broadening and baseline corrected using a polynomial function. Peak areas of metabolites were obtained by fitting mixed Lorentzian-Gaussian line shapes to ^13^C sum spectra (i.e., sum of individual ^13^C spectra acquired post DHA administration). We do not include ^2^H NMR spectra of the glucose in our analysis, as we achieved unexpectedly high enrichments of both ^13^C and ^2^H. Excessive ^13^C enrichment rendered the ^2^H spectra unusable due to overlap between the peaks and adjacent ^13^C j-coupled satellites.

### 4.4. Gas Chromatography—Mass Spectrometry

#### 4.4.1. Perfusate Extraction

Protein in 1 mL of perfusate was precipitated by adding 100 μL of 70% (*v*/*v*) perchloric acid. Samples were centrifuged. 1 mL of supernatant was collected and pH adjusted to 7 using 5 M KOH. The samples were centrifuged to pellet out potassium perchlorate salt before collecting 1 mL of supernatant. The supernatant was dried in a lyophilizer (Thermo Fisher Scientific, Waltham, MA, USA). The dried solution was reconstituted in 100 μL of 50:50 Acetonitrile water and then allowed to sit in a −20 °C freezer for at least 2 h before centrifugation and collection of 80 μL of the resulting supernatant. All centrifugation steps were carried out at 4 °C and 10,000× *g* for 30 min.

#### 4.4.2. Liver Extraction

Freeze clamped perfused liver samples were stored at −80 °C prior to analysis. 110 +/− 10 mg of perfused livers were transferred into conical bottom microcentrifuge tubes with screw caps. 1.0 mm zirconium oxide homogenization beads were added to each microcentrifuge tube along with 1 mL of cold degassed acetonitrile:isopropanol:water (3:3:2 *v*/*v*/*v*) solvent mixture. Samples were homogenized with a bead homogenizer (Fastprep-24, M.P. Biomedicals, Irvine, CA) for 3 × 20 s cycles, and were cooled on ice for 5 min between each cycle. The samples were then centrifuged at 10,000× *g* at 4 °C for 30 min. 900 μL of supernatant was recovered and lyophilized (Thermo FisherScientific, Waltham, MA, USA). The dried precipitate was reconstituted in 150 μL of acetonitrile:water (1:1 *v*/*v*) mixture, followed by incubation at −20 °C for at least 2 h. Reagents, unless otherwise specified, were purchased from Fisher Scientific, Waltham, MA, USA.

#### 4.4.3. Methoxyimino Penta-Trimethyl Silyl Derivatization

1 μL of extracted perfusate for *db/db* perfused livers and 25 μL of extracted perfusate for BLKS livers (perfused with metformin or glucagon) were loaded into 0.5 mL V-vials (Wheaton glass, Millville, NJ, USA) along with 400 ng of 2-hydroxyisovalerate and 400 ng of norleucine (added from stock solutions). These samples were dried under a gentle stream of air. The dried samples were then reconstituted in 50 μL of methoxyamine HCL in pyridine (Thermo Fisher Scientific, Waltham, MA, USA) and agitated using a microstirbar (vFisher Scientific, NJ, USA) for 1.5 h at 30 °C. 50 μL of N-Methyl-N-trimethylsilyl-trifluoroacetamide (MSTFA, CAS# 24589-78-4, Sigma Aldrich, St. Louis, MO, USA) was added to the reaction mixture and the reaction mixture was allowed to react for 30 min at 37 °C.

#### 4.4.4. MTBSTFA Derivatization

For analysis of amino acids and TCA cycle intermediates, ~8 mg of liver extract was dried down by airstream and was reconstituted in 50 uL of methoxyamine HCL in pyridine (Thermo Fisher Scientific, Waltham, MA, USA) for 1.5 h at 30 °C, while stirred by microstirbar. Then 50 μL of dimethyl tert butyl silyl trifluoroacetamide (MTBSTFA; ProteoSpec MTBSTFA w/1% TBDCMS, Ricca Chemical Company, Arlington, TX, USA) was added and allowed to react at 70 °C for 30 min while stirring. After derivatization, 80 μL of the sample was transferred to a GC vial for analysis by GC-MS.

#### 4.4.5. Aldonitrile Pentapropionate Derivatization

Using the same amount of extract as the methoxyimino penta-trimethyl silyl Derivatization for each sample group, the samples were dried in 0.5 mL V-vials and reconstituted in 50 μL of 20 mg/mL hydroxylamine hydrochloride (CAS# 5470-11-1; Acros Organics, Fair Lawn, NJ, USA) in pyridine (CAS#25104, Thermo Fisher Scientific, Waltham, MA, USA) and agitated with a microstirbar at 90 °C for 1.5 h. 100 μL of propionic anhydride (CAS# 240311-50G; Sigma Aldrich, St. Louis, MO, USA) was then added and samples were agitated with a microstirbar at 70 °C for 30 min. The samples were dried down by a gentle stream of air and then reconstituted in 100 μL of ethyl acetate for GC-MS analysis.

#### 4.4.6. GC-MS Method

After the appropriate derivatization reaction (MSTFA, MTBSTFA or Aldonitrile pentapropionate) was complete, 5 μL of each sample was pooled together to make a perfusate “pooled sample” and 80 μL of each individual sample was loaded into a 150 μL glass insert (CAS#13-622-207, Thermo Fisher scientific, Waltham, MA, USA) inside of a GC vial with a 9 mm PTFE/red rubber septum (CAS#C4000-30, Thermo Fisher Scientific, Waltham, MA, USA). The samples were loaded onto the AI/AS 1300 autosampler to await auto injection into the Trace1310 Gas chromatograph system (Thermo Fisher Scientific, Waltham, MA, USA). Samples of 1 μL volume were injected into the injection port with a splitless single taper liner at 250 °C and a splitless time of 60 s and column flow at 1 mL/min with a 30 m RTX-5MS integra Guard column (Crossbond 5% diphenyl/95% dimethyl polysiloxane CAT# 12623-127, Restek, Bellefonte, PA, USA) and 10 m guard column.

For MSTFA derivatized samples, the oven was held at 60 °C for 1 min, followed by a 10 °C/min ramp up to 325 °C followed by a 5 min bake out. For MTBSTFS derivatized samples, the following parameters were used: 60 °C hold time = 1 min, ramp of 15 °C/min up to 320 °C followed by a 5 min bakeout at 320 °C. Aldonitrile pentapropionate derivatized samples were run on the same GC-MS instrument with these parameters: start temperature of 80 °C with a hold time = 1 min, 20 °C per minute ramp up to 280 °C, followed by a 5 min bakeout at 280 °C.

For all samples, the transfer line was maintained at 280 °C and the ion source was maintained at 230 °C. The solvent delay on the mass spectrometer was set to 10 min. A *m*/*z* filter of 40–600 mass-to-charge ratio was used and individual metabolites were identified and quantified by area using their known quantification ions as in Appendix A. Concentrations were assessed against an eight-point standard curve that ranged from 50 ng to 1600 ng containing the metabolites.

#### 4.4.7. Peak Integration

Peak areas were integrated using Xcalibur (version 4.1) batch processing with genesis peak fitting, a 5 s time interval, height cutoff at 5.5% of the peak with valley detection enabled, and a S/N cutoff threshold of 3.

#### 4.4.8. Isotopic Ratio and Fractional Enrichment

The isotopic ratio of each metabolite for its given quantitation ion was calculated as the ratio of intensity of each mass isotopomer (*m_i_*) to the sum of all mass isotopomers ($\sum_{0}^{6}m_{i}$. The isotopic ratio distribution was then corrected for natural abundance, as previously described using the Isotopomer Network Compartment Analysis (INCA) software [38,39].

### 4.5. Statistical Analysis

Normality of data was established using the Shapiro-Wilk test. Unless otherwise stated, statistical significance was established using one-way ANOVA with post-hoc analysis using Tukey HSD or Kruskal-Wallis test, as appropriate. *p*-values less than 0.05 were considered significant.

### 4.6. Software

All NMR data processing was carried out using Mestrenova (v12.0.4 or newer; Mestrelab, Spain). Statistical analysis was carried out using R (v3.6.0). Figures were prepared using in house python (v3.8) scripts (libraries utilized were numpy (v1.19.2) [40], pandas (v1.1.3), matplotlib (v3.3.2) [41], nmrglue (v0.8) [42]), and Inkscape (v0.92.2).

## Figures and Tables

**Figure 1 metabolites-11-00441-f001:**
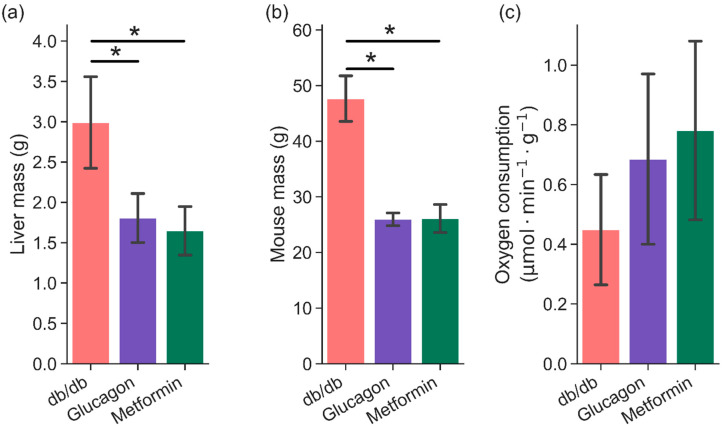
Comparison of (**a**) liver mass (wet weight), (**b**) mouse mass, and (**c**) hepatic oxygen consumption between three groups—*db/db* (n = 6), glucagon (n = 5), and metformin (n = 6). Hepatic oxygen consumption was measured during perfusion in approximately 10–15 min intervals. Average of all measurements are shown here. * indicates *p* < 0.05.

**Figure 2 metabolites-11-00441-f002:**
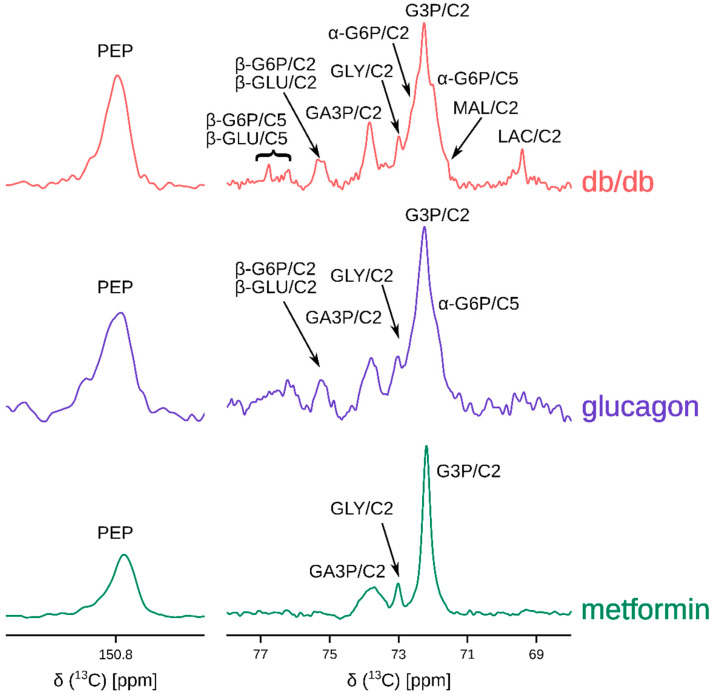
Representative ^13^C sum spectra obtained by adding individual transients of the pseudo-2D NMR acquisition post hyperpolarized [2-^13^C]dihydroxyacetone injection. Abbreviations are: PEP: phosphoenol pyruvate, GLU: glucose (α- and β-isomers), G6P: glucose-6-phosphate (α- and β-isomers), GA3P: glyceraldehyde-3-phosphate, GLY: glycerol, G3P: glycerol-3-phosphate, MAL: malate, and LAC: lactate. C2 and C5 represent carbons at positions 2 and 5. Metformin hydrochloride and glucagon concentrations in the perfusate were 10 mM and 50 nM. Appendix A provides a summary of metabolites observed for each group. Not all metabolites indicated were observed in every spectrum.

**Figure 3 metabolites-11-00441-f003:**
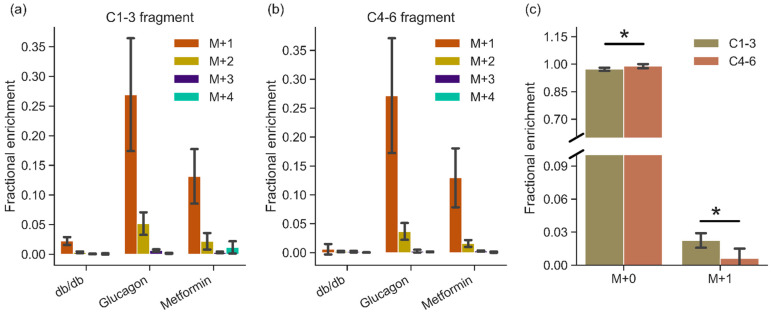
GC-MS analysis of glucose produced by the livers showing mass fragments containing (**a**) carbons 1–3 and (**b**) carbons 4–6 of glucose. Mass isotopomers are represented as M+0, M+1, M+2, M+3, and M+4 to indicate increasing mass of fragments. *m*/*z* values of fragments are listed in Appendix A. Natural abundance mass fragment (M+0) is not shown for clarity. (**c**) Comparison of M+0 and M+1 mass isotopomers between two halves of glucose (C1–3 and C4–6) produced by *db/db* livers (n = 6). Mass fragments are from carbons 1–3 and 4–6 in glucose post aldonitrile-pentapropionate derivatization reaction, as detailed in the methods. Error bars are standard deviation. n = 6 (*db/db*), 5 (glucagon), and 6 (metformin). ***** denotes statistical significance (*p* < 0.05; two tailed Student’s *T*-test).

**Figure 4 metabolites-11-00441-f004:**
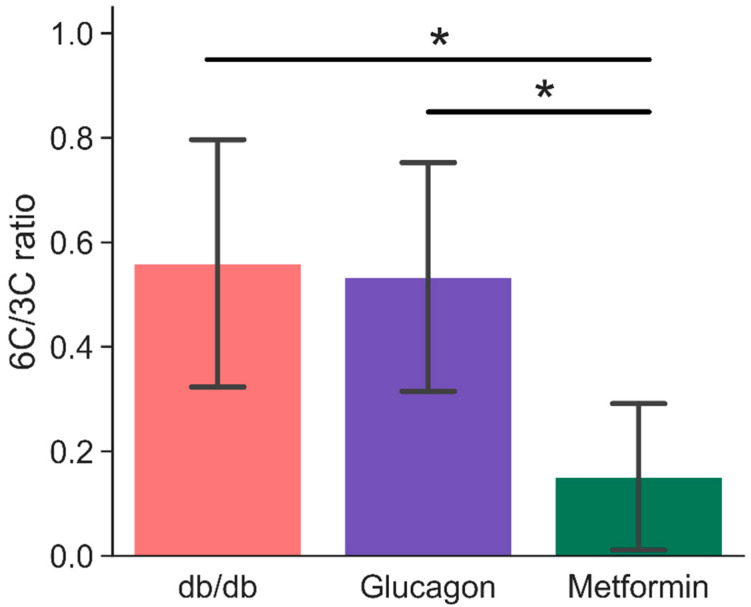
Ratio of signal intensities of six-carbon (glucose-6-phosphate and glucose—both α- and β-isomers) to three- and four-carbon (glyceraldehyde-3-phosphate, phosphoenol pyruvate, glycerol, glycerol-3-phosphate, lactate, and malate) resonances. Signal intensities were obtained by fitting a mixed Gaussian/Lorentzian shape to resonances in the sum spectra (shown in Figure 2). * indicates statistical significance (*p* < 0.05). Error bars are standard deviation. n = 6 (*db/db*), 5 (glucagon), and 6 (metformin). 6C/3C ratios were calculated based on observed set of metabolites for each spectrum. Appendix A provides a summary of metabolites observed for each group.

## Data Availability

All data presented in this publication have been submitted to the Metabolomics Workbench (www.metabolomicsworkbench.org/data, DataTrack ID: 2761).

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
