# Peer review of "Hyperpolarized Dihydroxyacetone Is a Sensitive Probe of Hepatic Gluconeogenic State"

_metabolites, 2021, doi:10.3390/metabo11070441_

Round 1

Reviewer 1 Report

Ragavan et al. have produced an elegant study using novel and technically sound biochemical modalities to permit further interpretation and context into hepatic metabolism in the setting of disease (diabetes) and pharmacological manipulation (acute glucagon and metformin). The main outcome variable was 6C/3C ratio as measured by the peak areas from the different six and three carbon metabolites post-HP DHA administration. This is commendable work that has the translatability to be moved to in-vivo models in the future. However, there are some concerns that I would like the authors to address:

  • Why is data from HP DHA in a healthy normal liver not provided in order to provide context into the effect size difference for 6C/3C ratio using HP DHA between healthy, disease, and pharmacological manipulation? It seems like this would make the study even more impactful.
  • Figure 2 assigns metabolite peaks to MAL/C2 and α-G6P/C2 that are not very convincing that they are truly peaks based on the representative image provided.
  • Please provide the mean ± SD for % polarization levels for HP DHA.
  • I am a little confused on why there were no differences in glucose production prior to HP DHA administration based on the lack of statistical differences in Fig S2. There is very little explanation of this data in the results and discussion. I would think that the db/db livers would have greater glucose production as measured in the perfusate or at least a similar amount as the glucagon treated liver. Please provide additional commentary in the results and discussion to enable the readers to see the whole picture more completely.
  • The statistical analysis section requires a bit more explanation. A t-test to compare three groups (diabetes, glucagon, metformin) via individual comparisons is not the most appropriate statistical model. A Kruskal-Wallis (non-parametric one-way ANOVA) could be used to test overall differences and then follow that up with post-hoc multiple comparisons would be more appropriate for this data.  Also, please explain in more depth how you are analyzing the isotopomer data statistically. It seems again like a two-way ANOVA model or at minimum when comparing within each label (i.e. M+1 mass isotopomer for example), a Kruskal-Wallis would be a better model.

Reviewer 2 Report

Please see the attache file for comment.

Reviewer 3 Report

The paper “Hyperpolarized Dihydroxyacetone is a Sensitive Probe of Hepatic Gluconeogenic State 3 y Mukundan Ragavan, Marc A. McLeod, Anthony G. Giacalone, and Matthew E. Merritt” is a short interesting study that clearly demonstrates how hyperpolarized DHA is sensitive to modulations of hepatic glucose output and proposes a simple metric to distinguish gluconeogenic and non-gluconeogenic livers. This is a prompt method directly applicable in vivo with current technologies.

Major comments:

Hyperpolarized (HP) 13C spectra are a function of several dynamic processes including transport phenomena (across the cellular and organelle membranes), reaction kinetics, and nuclear spin relaxation. I wonder if:

  1. spin−lattice relaxation experiments
  2. kinetic rate constants

could add further information due to their dependance on NMR measurements of dynamic processes.

Pag 2/16 lines 68-69 and Pag 6 lines 168-169 respectively report:

Hepatic oxygen consumption during the perfusions was not different between the BLKS livers (either group) and db/db livers as shown in Figure 1 (c).

Overall, substrate utilization and hepatic energetics were similar between the livers in the three groups, as evidenced by these results and the O2 consumptions.

The oxygen consumption is a relevant point because many time in the text is has been invoked at support of author’s hypotheses.

The similarities about oxygen consmption have to be better explained toghether with the very high errors values.

Minor comments:

pag 2 line 64 Figure 1 description has to be modified letter a) and b) shoul be inverted analogously in the figure 1 caption

In figure 2 b relative to glucagon perfusion, relations between metabolites and signals shoul be reported in NMR spectrum as for figures 2a and 2c

Round 2

Reviewer 1 Report

Thank you for your thoughtful responses. Excellent work.

Reviewer 2 Report

The authors have responded to the comments thoroughly.

Minor comments:

Line 248: Remove "."

Line 529: Remove a space.

Reviewer 3 Report

The paper 

Hyperpolarized Dihydroxyacetone is a Sensitive Probe of Hepatic Gluconeogenic State by Mukundan Ragavan , Marc A. McLeod , Anthony G. Giacalone , Matthew E. Merritt    can be accepted in the present form